# Differences in *Tibia* Shape in Organically Reared Chicken Lines Measured by Means of Geometric Morphometrics

**DOI:** 10.3390/ani11010101

**Published:** 2021-01-06

**Authors:** Domitilla Pulcini, David Meo Zilio, Francesco Cenci, Cesare Castellini, Monica Guarino Amato

**Affiliations:** 1Consiglio per la Ricerca in Agricoltura e l’Analisi dell’Economia Agraria, Centro di Ricerca Zootecnia e Acquacoltura, Via Salaria 31, 00015 Monterotondo, Italy; david.meozilio@crea.gov.it (D.M.Z.); f.cenciagronomo@libero.it (F.C.); monica.guarinoamato@crea.gov.it (M.G.A.); 2Department of Agricultural, Environmental and Food Science, Università degli Studi di Perugia, Borgo XX Giugno 74, 06124 Perugia, Italy; cesare.castellini@unipg.it

**Keywords:** bone shape, genetic lines, geometric morphometrics, organic, outdoor rearing, poultry

## Abstract

**Simple Summary:**

Organic poultry production should use only genetic lines and animals resistant to disease and well adapted to live outdoor, according to principles, rules, and requirements of organic farming systems. When broiler’s walking performance is reduced animals are not suitable for outdoor rearing. There is a straightforward relationship between bone health and growth rate in broilers. Body and breast weight play an important role in leg disorders. During the last decades, genetic selection has led to high producing broilers over the time. Unfortunately, fast growth may negatively influence correct leg development, reducing walking performance, and raising welfare issues. Leg abnormalities could represent a criterion for the choice of genetic lines suitable for organic production. A method for their early detection was developed in this study by means of Geometric Morphometrics (GM) that represents a tool for bone shape analysis and its correlation with walking capability. A valuable information emerged from the present study in relation to broiler intrinsic adaptability to organic production.

**Abstract:**

In the present study, the conformation of the *tibia* of seven genetic lines of broilers was analyzed by Geometric Morphometrics and correlated to carcass weight and walking ability. The used chicken genetic lines were classified as fast, medium, or slow growing and ranked for their walking ability. Six chicken types were reared in an organic farm and slaughtered at 81 days of age while one slow-growing and highly walking line (Naked Neck) was reared in a commercial farm and used as external reference for moving activity and growth speed. A mixed landmarks and semi-landmarks model was applied to the study of *tibia* shape. Results of this study showed that: (i) body weight gain was positively correlated to the curvature of the antero-posterior axis of the *tibia*; (ii) the shape of the *tibia* and the active walking behavior were significantly correlated; (iii) walking and not-walking genetic lines could be discriminated in relation to the overall shape of the *tibia*; (iv) a prevalence of static behavior was correlated to a more pronounced curvature of the antero-posterior axis of the *tibia*. Results of this study revealed that the walking genetic types have a more functional and natural *tibia* conformation. This easy morphologic method for evaluating *tibia* shape could help to characterize the adaptability of genotypes to organic and outdoor rearing.

## 1. Introduction

According to EU Organic Regulations (Reg. EC N° 834/2007 and 889/2008), organic poultry production should be based on resistant genotypes able to live in outdoor conditions and to favorably exploit outdoor runs for foraging and exploring. When walking ability is reduced, animals are not suitable for rearing in open air; walking ability depends on genotype, weight, age, feeding, and housing [1]. It has been demonstrated that fast-growing broilers, if reared in organic system, are genetically prone to lameness and leg weakness [2]. There is a straightforward relationship between bone quality and growth rate in broilers [3]. Body weight plays a decisive role in lameness [1]. Genetic selection produced very efficient broilers, but fast growth can influence the isometric growth, causing leg problems [4] and changes in leg bones biomechanics [5]. Slow-growing broilers show less leg problems or lameness [1], but, as they have worse feed conversion rates and lower commercial weights, are often avoided by farmers who prefer most medium/fast genetic types. It should be noted that heavy weight is closely correlated with rotated *tibia* [6] and with the lateral curvature of the upper *tibia* [7]. Rotated *tibiae* can be also the result of too early rapid growth rate and low activity [8], or a genetic issue, since it was evidenced also in slow growing genotypes [9].

In this complex framework, the identification of genetic lines showing the tendency to develop leg anomalies, even at a non-pathological stage, is pivotal, in order to prevent welfare and health issues, in particular for organic poultry production.

This study was part of a project (TIPIBIO) focusing on poultry adaptability to organic systems from different point of view such as behavior, welfare, and metabolic status. One specific object of the project was to define a plausible adaptability index for the different genotypes. Leg anomalies (better if rapidly diagnosed) should be included in that index, given their strict relation with walking ability,

Many studies on broiler welfare in relation to leg problems used a gait score method, e.g., the Bristol scoring system, assessing in vivo ability to walk (from 0 to 5) [10]. Such inspective, non-invasive methods do not take into consideration bone pathologies [9] or bone anomalies but are very rapid and easy. Nevertheless, the scoring system is not fully objective, depending on the operator.

The morphological approach is a more objective method used to detect bone anomalies, cause of lameness, and bone defects. To the best of our knowledge, previous studies relating fast growing and body weight to bone health and morphology used classic morphometrical approaches, such as the measure of bone length, weight, diameter [2,3,11], or analysis of radiographic images [12]. Most of previous studies considered the occurrence of skeletal anomalies and pathologies [11,13,14,15]. Significant differences were found in morphometric traits of tibio-tarsus in two different broiler strains (Ross and Lohman Dual) [2]. Rapid growth genotypes showed more *tibia* deformities, lack of mineralization, and reduced strength.

In the present study, morphological changes in the *tibia* of different poultry genetic lines (meat type) were analyzed by the Geometric Morphometrics (GM) [16,17], a tool for shape analysis able to quantify changes in overall bone morphology and their relationship with other variables (e.g., genetic lines and body weight). GM easily permits localization of morphological changes and information about the magnitude of variation can be extracted [18] and visualized through deformation grids and vectors [19].

The aims of the present study were: (1) To assess the effects of genetic line on the shape of *tibia* in seven commercial lines of chickens; (2) to investigate the relationship between walking behavior and shape of *tibia*; (3) to provide useful information for selection of genetic lines suitable for organic production.

## 2. Materials and Methods

A total of 105 right *tibiae* were analyzed in this study; 90 samples were obtained from six poultry lines, from fast- to slow-growing ones (Table 1), reared in the experimental farm of the University of Perugia (Italy), within the TIPIBIO project, as fully described in a companion study [20]. Ninety male chickens from six commercial lines, Aviagen Ranger Classic (RC), Ranger Gold (RG) and Rowan Ranger (RR), Hubbard CY Gen5xJA87 (CY) M22xJA87 (M), and RedJA (C), were reared in an organic farm and slaughtered at 81 days of age (the minimal slaughtering age required for organic production, in compliance with Regulation CE 889/2008). The growth rate classification of the different lines was provided by the breeding companies. As for Aviagen strains, a previous study confirmed RC as fast-growing, RG as medium-growing, and RR as slow-growing [21]. For Hubbard lines, no comparative studies were available, but the breeder company ranking was applied as well (i.e., CY and M medium/fast growing and C slow growing). The remaining 15 *tibiae* were taken from Naked Neck (NN) chickens reared in an organic farm in Center Italy (Jesi, AN). NN were used as a benchmark, as it is normally used for free-range production (Label Rouge).

Genetic lines have been so regrouped in Walking (W) and Not-Walking (NW) (Table 1), according to the results of a previous study [20], which characterized genetic lines depending on the time spent in active or static behaviors. Static behaviors were (1) rest (i.e., body in line with the ground, with erect head and open eyes) and (2) roost (i.e., be standing, no body movement, erect or relaxed head with open eyes). Walking (i.e., moving more than three steps in one direction with upright head) was classified as an active behavior. Carcass weight and breast yield were recorded (Table 2). Right *tibiae* were excised, fleshed out, and boiled in order to remove residual meat, ligaments, and tendons. The correlation of the measured traits with both carcass and breast yield was calculated, but being very similar, we decide to report only that with carcass weight.

All adopted procedures were in accordance with the EU legal framework relating to the protection of animals used for scientific purposes (Directive 2010/63/EU).

Each *tibia* (*n* = 105) was photographed in lateral aspect by a high-resolution digital camera (13 real MP) set on a tripod. The focal distance between the camera and the *tibia*e was 40 cm. For each *tibia*, length (measured between the ends of proximal and distal epiphyses) and measure of arch width (BH), a proxy of antero-posterior curvature [11,22] were collected using TpsDig2.0 [23], as illustrated in Figure 1. Landmarks (4) and semi-landmarks (16) were digitalized on each image using TpsDig2.0 (Figure 1). Landmarks were digitalized on the proximal anterior (1) and posterior (2) epiphyses and on the distal anterior (3) and posterior (4) epiphyses. As no other homologous landmarks should be identified on the *tibia*, in order to ensure shape coverage of the entire bone, two outline curves were recorded on each *tibia*. The first beginning from landmark 1 and ending at landmark 2 and the other beginning from landmark 3 and ending at landmark 4. On each of these two outlines, eight equally spaced semi-landmarks [18,24,25] were automatically digitalized, along each curve, using TpsDig2.0. Semi-landmarks could slide iteratively along the outlines curve using the spline relaxation procedure algorithm of Bookstein [26,27]. After relaxation, semi-landmarks can be treated in multivariate analyses as homologous points [26,27]. Landmarks and sliding landmarks were converted into shape coordinates using Procrustes superimposition [28], removing information about location and orientation from the raw coordinates and standardizing each specimen to a unit centroid size (CS—square root of the summed squared Euclidean distances from each landmark to the specimen’s centroid). Residuals were analyzed using the thin-plate spline (TPS) interpolating function [19], producing principal warps.

Morphometric software of the TPS series are freely available (http://life.bio.sunysb.edu/morph/).

Differences in carcass weight, *tibia* length, and *tibia* arch width among genetic lines were tested by the one-way analysis of variance (ANOVA, Welch’ test for unequal variances) with Tukey post-hoc pairwise test. Arch width and *tibia* length were significantly correlated (Spearman *r* = 0.5, *p* < 0.01), so ANOVA for arch width differences in genetic lines was performed on least square regression residuals.

To display *tibia* shape variation, principal component analysis (PCA) was performed on the covariance matrix. Spearman’s rank correlation coefficient was calculated between PC1 and PC2 scores and carcass weight. Individuals were distinguished in the plot according to their walking aptitude (as codified in Table 1).

Discriminant function analysis (DFA) was performed to separate *a priori* known groups in the data, providing an ordination that maximized the separation of the group means relative to the variation within groups. Genetic lines were separated into two categories: Walking and not-walking (Table 1). DFA procedure carries out a leave-one-out cross-validation to assess the reliability of classification [29].

Statistical tests were performed using PAST. PCA and DFA were performed using the software MorphoJ.

The pattern of covariation between the shape of the *tibia* and behavior (active or static behaviors, as measured by [20], was analyzed using partial least squares (PLS) analysis [18,30] only on the six genetic lines reared in the experimental section of the University of Perugia [20]. For details and computational aspects, see [19,30,31,32].

## 3. Results and Discussion

### 3.1. General Results

Average carcass weight, length of the tibiae, and width of the arch (BH) for each genotype are shown in Table 2. The six lines showed significant differences in final carcass weight (F = 35.5, *p* < 0.01): The reference group (NN) was different from all the other groups and showed the lowest carcass weight. These results are in line with the organic rearing conditions of NN. Two not-walking lines (RC and CY) showed higher carcass weight values, significantly different from walking ones (RG, RR, and C). Hubbard M22xJA87 (M), even though classified as a not-walking line, was not statistically different from W lines concerning the final carcass weight. The length of the tibia was significantly different among groups (F = 22.6, *p* < 0.01). The highest length of the tibia was measured in M and C strains, while the lowest in the reference strain (NN), followed by the RC. However, post-hoc Tuckey-pairwise comparisons were significant only between NN and all the other groups probably due to breed effect and different rearing conditions as well.

Arch width, which could be considered a proxy of the curvature of the tibia, was significantly different among groups (F = 3.9; *p* < 0.01). The reference group (NN) was significantly different from all other groups, with the lowest curvature. A cluster including walking genetic lines (RG, RR, and C) showed intermediate values, while higher values were recorded for not-walking ones (RC, CY, and M).

The arch width (BH) was correlated with antero-posterior curvature and chicken growth. Shim et al. (2012) found that fast-growing lines show higher risk of tibia breakage due to lower bone density, confirmed by a decrease in mechanical strength and ash content, when compared to slow-growing ones. In a previous study, authors observed a higher antero-posterior curvature in chickens affected by varus limb [22]. Other authors reported a tendence to higher antero-posterior curvature in broilers with leg disorders [11]. In our study, antero-posterior curvature simply measured as the arch width (BH) was not enough to discriminate between walking and not-walking groups, rising the need of a more comprehensive multiparametric indicator (e.g., bone shape).

### 3.2. Shape Analysis

The first two axes of PCA accounted for 69% of total variance (Figure 2). Even if the two groups (W: Walking; NW: Not-walking) were partially overlapped, observations related to the W group were mostly located in the positive part of PC1 (54% of variance explained), while NW observations were in the negative portion. The reference group (NN) was mainly distributed in the very right side of the axis PC1. Along PC2 (15% of variance explained) the two categories were almost completely overlapped. Negative scores on PC1 corresponded to a more evident curvature of the tibia antero-posterior axis, compared to the straighter configuration of this bone in specimens positioned in the positive region of the same axis. Scores along PC1 were negatively correlated with carcass weight (Spearman *r* = *−*0.44, *p* < 0.05), thus weight increase was correlated to a more evident curvature of the tibia.

Walking (W) and not-walking (NW) genetic lines were significantly discriminated by discriminant analysis (T2 = 106.5; *p* < 0.05). Groups were slightly overlapped and classification was very reliable (Figure 3; Table 3). The percentage of correct classification to *a priori* defined groups after cross-validation was up to 80%. The highest error occurred in the correct attribution of individuals to not-walking class (16.7%).

The shape of *tibia* and chicken behavior (active/static) (Figure 4) were significantly correlated (*r* = 0.40; *p* < 0.05). Genetic types classified as walking had active behaviors ranging between 0% and 11%; in those classified as not-walking, active behaviors ranged between 24% and 37% and the highest values was observed for RR. A prevalence of static behavior was correlated to a more pronounced curvature of the antero-posterior axis of the *tibia*.

These results are consistent with the above-mentioned studies evidencing a strong relationship between leg deformities, lameness, and consequently walking activity. A positive correlation between time spent lying and lameness was found in broilers [33] and most cases of lameness can be attributed to leg anomalies. One of the main causes of leg diseases in commercial broilers is the selection for rapid growth and high body weight, which affect chick skeletal development and conformation, causing bone anomalies [34]. Bone anomalies, therefore, present a certain degree of heritability [35], making some genetic lines more prone to develop lameness than others [36], even if the relationship between leg disorders and genetic factors is not yet clarified. Morphological changes in fast-growing broilers, such as the increased weight of breast muscles, may cause changes in bone structure and morphology, moving forward the center of gravity of the animal [34]. High incidence of leg pathologies is related to growth rate and breast muscles yield and negatively affected the normal chicken anatomy and physiology [37]. Broilers affected by valgus-varus disease had impaired walking ability and lower body weight, probably due to the difficulty of drinking and eating [11]. In contrast with these last findings, in the present study, carcass weight was positively correlated to antero-posterior curvature of *tibia*. Anyway, results are not comparable, as all the chickens analyzed are healthy and apparently free of leg diseases. In support of this claim, [38] found that fast-growing broiler are more subjected to tibio-tarsus mineralization disorders and bone fractures, being the bones overcharged for high body weight [39,40]. Bone quality (i.e., mineralization and bone density) of slow-growing lines was higher than that of fast-growing ones [3,41,42,43]. The rigidity and consequently the bending resistance, of tibiotarsus in the Lohman Dual slow-growing line was greater than in Ross 308, a highly selected fast-growing line [2], confirming that skeleton stability is affected by weight gain.

A correlation between growing rate and behavioral repertoire as well as welfare conditions have been evidenced [44]: Slow-growing lines showed lower gait scores than fast-growing and spent more time in active behaviors, assuming the same diet and stocking density.

## 4. Conclusions

The correlation between growth rate and bone health in broilers has been previously documented [3]. Leg disorders are serious concern for poultry industry and cost over $100 million per year to producers [45]. In the present study, a Geometric Morphometric approach was applied to the analysis of shape variation of the *tibia* in broiler lines with different growth rates. Relationship between *tibia* shape and walking attitude was investigated. The conclusions are:
*Tibia* shape could represent a reference/starting point for studies on chicken walking behavior. Statistical analysis evidenced a clear and straightforward correlation between the curvature of this bone and the tendency of the animal to spend more time in static activities, such as resting and roosting;Degree of curvature of the *tibia* is positively related to carcass weight and to growth rate. Fast-growing genotypes showed a more pronounced curvature of the *tibia*;Ranger Gold, Rowan Ranger, and RedJA, previously classified as walking poultry lines [20], showed a less curved *tibia*, similar to that of the reference line (the slow-growing Naked Neck).

Besides, the results of this study suggest that walking lines are associated to favorable *tibia* shape and characteristics. The easy and cheap (mainly based on open-source software) method used for *tibia* shape and curvature definition is ideal for defining an early index of the adaptability of specific genotypes to organic and outdoor rearing.

All animals were apparently healthy and none of them showed evident leg pathologies all over the experiment, but it is rational that heavier chickens, characterized by a curved *tibia*, could be more prone to deformities or pathologies and more exposed to lameness risk [2,3].

Further studies are necessary to validate these results by the inclusion of left leg in the analysis (right is supposed to be more sensitive to deformation). A larger number of individuals and genetic lines included in the study could help to have a more faithful representation of the reality and would balance anatomic markers also.

## Figures and Tables

**Figure 1 animals-11-00101-f001:**
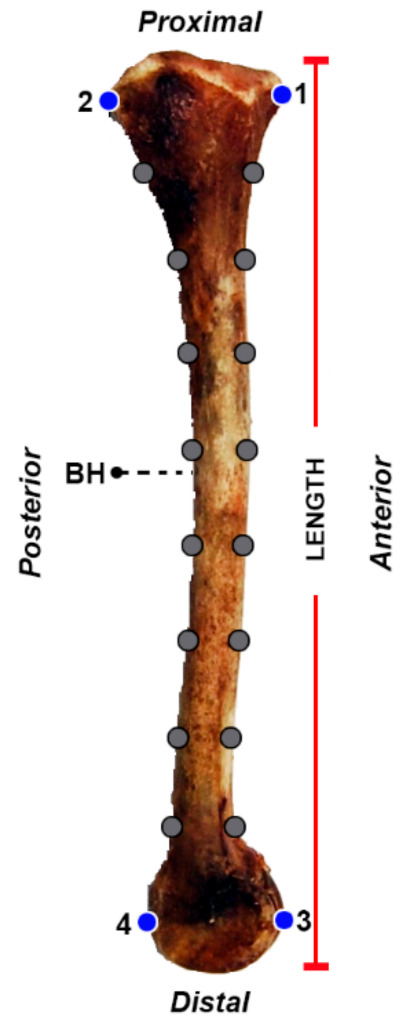
Landmarks (blue circles) and semi-landmarks (grey circles) collected on the *tibia*: (1) Proximal anterior and (2) proximal posterior epiphyses; (3) distal anterior and (4) distal posterior epiphyses. Biometrics collected along the *tibia*: Length and maximum width of the arch described by the bone (BH).

**Figure 2 animals-11-00101-f002:**
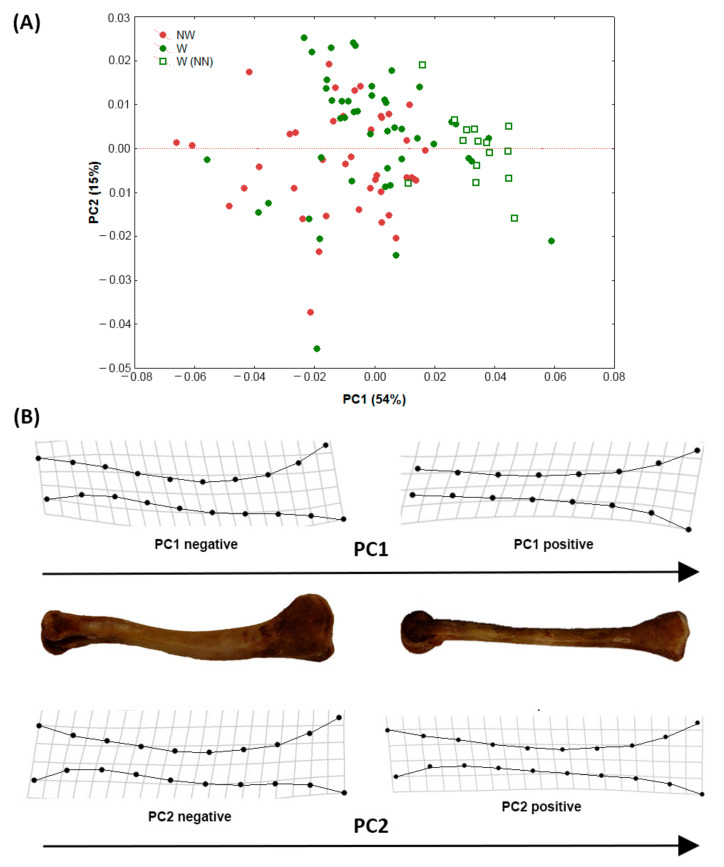
(**A**) Principal component analysis plot. Grouping variable: Walking (W)/not-walking (NW)/walking naked neck (W-NN). (**B**) Shape variation along PC1 and PC2 was represented by splines relative to positive and negative extremes of the axes. For shape variation along PC1, tibiae of two extreme individuals were reported.

**Figure 3 animals-11-00101-f003:**
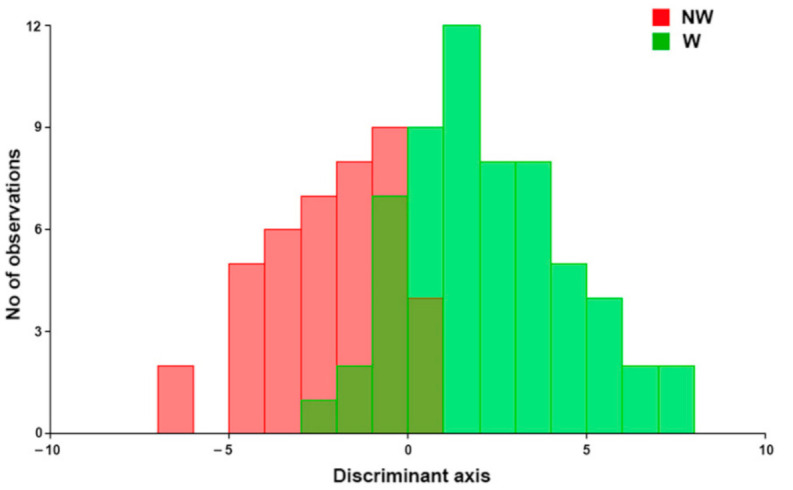
Discriminant analysis histogram. Grouping variable: Walking (W)/not-walking (NW).

**Figure 4 animals-11-00101-f004:**
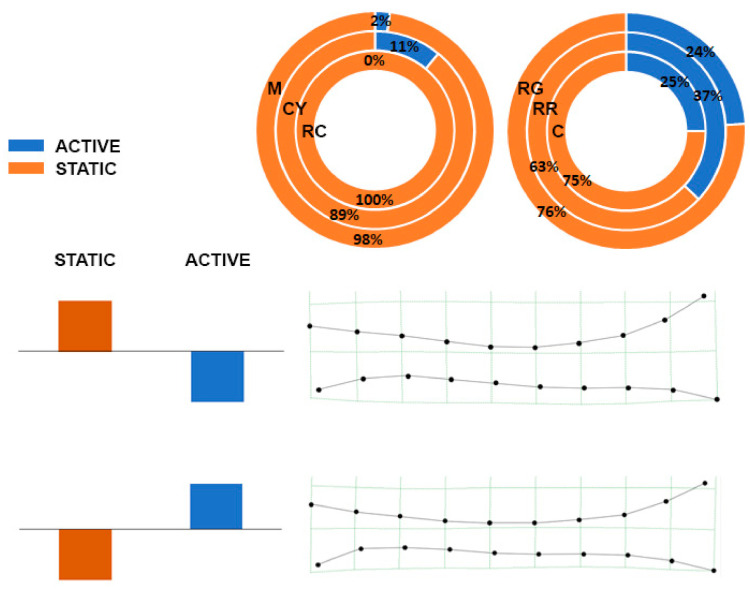
Partial least squares (PLS) showing the morphological relationship between the *tibia* and the walking/resting behavior described as percentage of time spent in two main activities (Walking W and Not Walking—NW). Percentages for each genotype are represented in pie charts. The splines depict *tibia* shape configuration corresponding to opposite patterns of behavior.

**Table 1 animals-11-00101-t001:** Genetic lines analyzed, walking behavior (W: Walking; NW: Not-Walking), and growth rate.

Genetic Line	Acronym	Walking Behavior	Growth Rate
Aviagen Ranger Classic	RC	NW	Fast
Aviagen Ranger Gold	RG	W	Medium
Aviagen Rowan Ranger	RR	W	Slow
Hubbard CY Gen5xJA87	CY	NW	Medium/Fast
Hubbard M22xJA87	M	NW	Medium/Fast
Hubbard RedJA	C	W	Slow
Hubbard RedJA Naked Neck	NN	W	Slow

**Table 2 animals-11-00101-t002:** Carcass weight (mean ± S.D.), length of the tibia (mean ± S.D.), and width of the arch (BH, mean ± S.D.) of the six genetic lines and of naked neck (NN). Different letters indicate significant differences among treatments (Tuckey test; *p* < 0.05); similar letters depict no significant differences.

Acronym	Carcass Weight (g)	*Tibia* Length (cm)	BH (cm)
RC	3616.0 ± 54.4 ^b^	13.4 ± 0.4 ^b^	1.3 ± 0.2 ^ab^
CY	3487.7 ± 117.6 ^b^	13.9 ± 1.3 ^b^	1.3 ± 0.2 ^ab^
M	3213.3 ± 121.2 ^c^	14.2 ± 0.8 ^b^	1.4 ± 0.1 ^a^
RG	3179.3 ± 71.7 ^c^	14.1 ± 0.5 ^b^	1.2 ± 0.1 ^bc^
RR	2859.33 ± 115.0 ^c^	13.6 ± 0.8 ^b^	1.1 ± 0.2 ^c^
C	2930.7 ± 77.2 ^c^	14.2 ± 0.6 ^b^	1.1 ± 0.2 ^c^
NN	2431.6 ±61.7 ^a^	12.0 ± 0.6 ^a^	0.9 ± 0.1 ^d^

**Table 3 animals-11-00101-t003:** Discriminant analysis classification/misclassification scores.

True	Allocated to		Total	%
	W	NW		
W	37	4	41	90.2
NW	10	50	60	83.3

## Data Availability

The data presented in this study are available on request from the corresponding author.

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
