# Peer review of "Differences in Tibia Shape in Organically Reared Chicken Lines Measured by Means of Geometric Morphometrics"

_animals, 2021, doi:10.3390/ani11010101_

Round 1

Reviewer 1 Report

This study is complicated. The use of the Naked Neck as a reference population is complicated by the fact that it was reared separately from the experimental lines. This, at least potentially, leads to important differences in the environments faced by the experimental birds. A better direct correlation between tibia shape and walking behaviors would also be ideal, but would require a within-strain analysis rather than a between-strain analysis. Such an analysis would strengthen the conclusions.

Author Response

Reviewer 1

  • This study is complicated. The use of the Naked Neck as a reference population is complicated by the fact that it was reared separately from the experimental lines. This, at least potentially, leads to important differences in the environments faced by the experimental birds.
  • A better direct correlation between tibia shape and walking behaviors would also be ideal but would require a within-strain analysis rather than a between-strain analysis. Such an analysis would strengthen the conclusions.

In our opinion, the presence of a reference line, known for its high adaptability to outdoor rearing and its walking aptitude, is very useful, providing a baseline for tibia shape. It is important to stress out that Naked Neck samples were used only in shape analysis and was not considered in the Partial Least Square Analysis (as stressed in the manuscript).

In this study, we applied two different statistical techniques to the study of shape:

  1. Principal Component Analysis, as an exploratory multivariate technique, often used in shape analysis, also known with the term Relative Warp Analysis. As the shape dataset is very complex, a technique producing lower-dimensional data, while preserving as much of the data's variation as possible, is necessary. It must by noted that PCA does not take into account any difference among groups. In this case, the presence of a reference sample (Naked Neck) does not change the general results: relative distances among observations in the shape space are not altered. Including NN in the analysis, lead to an increase of variation explained by the first principal component (PC1) from 50% to 54% and to a decrease in variation explained by PC2 from 19 to 15%. Therefore, the overall variance explained by the two-dimensional shape space was the same, but the variance of the projected data was maximized including the NN sample.
  2. Discriminant Analysis, which starts with an initially defined grouping (walking and not walking) of objects and tries to determine to which extent a set of quantitative descriptors (shape data) can efficiently explain this grouping. It is closely related to principal component analysis, as they both look for linear combinations of variables which best explain the data, but DA explicitly attempts to model the difference between the classes of data.

In shape analysis, such techniques are often used in combination, the first as an exploratory technique, the second as a tool to test the effective validity of grouping observations in a priori defined classes on the base of shape data.

Partial Least Square Analysis was chosen as a statistical tool in order to find the relations between two matrices (the matrix of shape data and the matrix of behavioral data). The PLS model try to find the multidimensional direction in the “shape” space that explains the maximum multidimensional variance direction in the “behavior” space. All these statistical techniques have been successfully applied to shape analysis and are implemented in several softwares, offering the opportunity to perform statistical tests and, at the same time, to visualize shape variation.

We did not completely understand the reviewer concerns. A between-group analysis was only performed in the case of discriminant analysis, but this technique performs a cross-validation based on permutations, thus testing within-group variability.

  • Extensive English Editing

Authors made a serious effort in order to improve the English of the manuscript.

Reviewer 2 Report

Comments to the Authors of manuscript number: animals-1062008 entitled “Differences in tibia shape in organically reared chicken lines measured by means of Geometric Morphometrics”.

Authors have presented the relationship between the shape of tibia and walking behavior in seven commercial lines of chickens, which can be useful in organic production. For this purpose They use geometric analysis. It is very interesting study. The introduction showed in correct manner the problem in poultry production. The manuscript is written properly creating a logical whole.

The manuscript should be corrected in some points.

  1. L 24, L 28, L80 and ohers.. small letters in full term
  2. L 78 – reference add
  3. L 93 the sentence should be finished properly
  4. L 96 Why broiler chickens were slaughtered at 81 days of life. It should be explain correctly, because common period include 43 days.
  5. L 91 L93 90 samples and 90 chickens, does it means that one bone from one chicken?

Do come next 15 bones from 15 NN chickens? It should be described in details.

  1. Were used left or right tibiae?
  2. L 110 was the bone deboned?? What it means?
  3. Is not needed the permission of local ethical committee?
  4. L 103 the reference should be added
  5. There is no information how chickens were held? It means how many chickens were in one cage and what the area of the cage was?
  6. there is no information about the sex of the chickens. It should be added.
  7. There is no information about sexual maturity and behavior. It should be described. Were encounters observed between chickens? Or how fodder and water were located? Were chickens forced to walk to eat or they have fodder very closely? All these points should be described.
  8. there is no information about terrain obstacles. If they were in chickens areas they should be described.
  9. In general, the condition under which the rearing was performed should be described in details.

Author Response

Reviewer 2

The manuscript should be corrected in some points.

  • L 24, L 28, L80 and others. Small letters in full term

Capital letters were used for the Geometric Morphometrics tool, because this is a clearly defined ad references method for shape analysis.

  • L 78 – reference add

The reference was added, as suggested by the reviewer.

  • L 93 the sentence should be finished properly

The sentence was modified, with the attempt to clarify that this study is a companion of Cartoni Mancinelli et al., see reference [20].

  • L 96 Why broiler chickens were slaughtered at 81 days of life. It should be explained correctly, because common period include 43 days.

Chickens were slaughtered at 81 days because they were reared according to organic rules (art. 12 of Reg. 889/2008).

  • L 91 L93 90 samples and 90 chickens, does it means that one bone from one chicken?

Only the right tibia was used for each specimen, and it was clarified in the text. In future studies, possible asymmetries between left and right tibia shape should be analyzed.

  • Do come next 15 bones from 15 NN chickens? It should be described in details.

At L 112, we added a statement about the origin of the remaining 15 bones.

  • Were used left or right tibiae?

We used right tibiae, it was clarified in the text.

  • L 110 was the bone deboned?? What it means?

According to the reviewer suggestion, the right term (i.e., fleshed out) was used.

  • Is not needed the permission of local ethical committee?

We used this statement at Lines 132-133: “All adopted procedures were in accordance with the EU legal framework relating to the protection of animals used for scientific purposes (Directive 2010/63/EU).” It should be sufficient in this case study.

  • L 103 the reference should be added.

The proper reference was added.

  • There is no information how chickens were held? It means how many chickens were in one cage and what the area of the cage was?
  • There is no information about the sex of the chickens. It should be added.
  • There is no information about sexual maturity and behavior. It should be described. Were encounters observed between chickens? Or how fodder and water were located? Were chickens forced to walk to eat or they have fodder very closely? All these points should be described.
  • There is no information about terrain obstacles. If they were in chickens areas they should be described.
  • In general, the condition under which the rearing was performed should be described in details.

Answers to question from 11 to 15 were grouped together. The present study is a companion of Cartoni Mancinelli et al., see reference [20]. In that study, the rearing conditions are fully described, we just summarized them in the text, to not incur self-plagiarism. The farm was organic, all the animals had the same diet, veterinary care, inside and outside spaces according the Regulation CE 889/2008.

Round 2

Reviewer 1 Report

This version is much improved, with a better explanation of much of the background information. The English is still weak in some places, but is now much more understandable.

Reviewer 2 Report

The manuscript can be accepted in this form